# Implementation of FAIR Practices in Computational Metabolomics Workflows—A Case Study

**DOI:** 10.3390/metabo14020118

**Published:** 2024-02-10

**Authors:** Mahnoor Zulfiqar, Michael R. Crusoe, Birgitta König-Ries, Christoph Steinbeck, Kristian Peters, Luiz Gadelha

**Affiliations:** 1Institute for Inorganic and Analytical Chemistry, Friedrich Schiller University Jena, 07743 Jena, Germany; mahnoor.zulfiqar@uni-jena.de; 2Cluster of Excellence Balance of the Microverse, Friedrich Schiller University Jena, 07743 Jena, Germany; birgitta.koenig-ries@uni-jena.de; 3ELIXIR (The European Life-Sciences Infrastructure for Biological Information) Germany, Institute of Bio- and Geosciences (IBG-5)—Computational Metagenomics, Forschungszentrum Jülich GmbH, 52428 Jülich, Germany; mrc@commonwl.org; 4Institute for Informatics, Friedrich Schiller University Jena, 07743 Jena, Germany; 5iDiv—German Centre for Integrative Biodiversity Research, Halle-Jena-Leipzig, 04103 Leipzig, Germany; kristian.peters@ipb-halle.de; 6Geobotany and Botanical Gardens, Martin-Luther University of Halle-Wittenberg, 06108 Halle, Germany; 7Leibniz Institute of Plant Biochemistry, 06120 Halle, Germany; 8German Cancer Research Center (DKFZ), 69120 Heidelberg, Germany

**Keywords:** workflow, FAIR, cheminformatics, metabolomics, CWL, CommonWL, Workflow RO-Crate, Docker, WorkflowHub, Bioschemas

## Abstract

Scientific workflows facilitate the automation of data analysis tasks by integrating various software and tools executed in a particular order. To enable transparency and reusability in workflows, it is essential to implement the FAIR principles. Here, we describe our experiences implementing the FAIR principles for metabolomics workflows using the Metabolome Annotation Workflow (MAW) as a case study. MAW is specified using the Common Workflow Language (CWL), allowing for the subsequent execution of the workflow on different workflow engines. MAW is registered using a CWL description on WorkflowHub. During the submission process on WorkflowHub, a CWL description is used for packaging MAW using the Workflow RO-Crate profile, which includes metadata in Bioschemas. Researchers can use this narrative discussion as a guideline to commence using FAIR practices for their bioinformatics or cheminformatics workflows while incorporating necessary amendments specific to their research area.

## 1. Introduction

A computational workflow represents a coherent chain of interrelated computational activities that process input data to produce output data, altogether depicted as one research object. Workflows have gained momentum and are continuously expanding in different research fields, such as biological data analysis and machine learning [1]. To administer the execution and monitoring of different computational activities in a workflow, many workflow management systems (WfMSs) have been developed [2,3], leading to different, incompatible ways to implement workflows, including control over the dependencies and order of execution, sometimes with a graphical user interface (GUI) [4]. However, most of these workflows remain poorly documented, and the increasing number of such systems leads to a lack of standardisation without using a common format for defining workflows. This hinders the reusability of the workflows for different purposes (such as applying the workflow for a different dataset) and the reproducibility of results from the same dataset [5]. 

The FAIR (findable, accessible, interoperable, reusable) principles, initially introduced in 2016 [6], were formulated with the primary objective of promoting the reuse of scientific data and data management [7] and were later extended to research software and workflows with the goal of uniformity and reusability. Since the advent of FAIR principles, different tools have been developed to enable uniformity in these workflows [8,9,10,11]. Generally, the initial workflow design plans are focused on the research goal rather than the FAIRification of the workflow; hence, the FAIR guidelines are not defined early in the process. Such workflows remain “unmanaged” [12], which leads to inconsistency and a lack of reusability across different systems. 

Many standardised practices that support the FAIR principles are emerging, starting from initiating the workflow development procedure [13]. One of the most significant achievements of the FAIRification of workflows is the launch of WorkflowHub, which serves as a workflow registry and explicitly supports the FAIR principles [14]. WorkflowHub is a platform that enables findability and accessibility by allowing the uploading of workflows using standardised metadata. These workflows not already written in the Common Workflow Language (CWL) are described with a “non-executable” abstract version of the CWL [12], along with the native WfMS, to provide a uniform definition across the Hub. 

The CWL standards project is a FAIR-aware initiative for describing and sharing workflows in a WfMS-independent way. It uses standards and machine-readable formats for workflow definition. With the CWL, users can easily share and reuse the workflows across different platforms and domains. Some CWL-aware workflow executors (like the CWL reference implementation, *cwltool*) also consistently collect relational data or provenance between different research artefacts of the workflows. The flow of activities between different modules during workflow execution and the associated data and metadata can be packaged as one entity to facilitate reusability. The Research Object Crate community [15] initiated the efforts in the Workflow RO-Crate profile, which captures the metadata associated with the workflow in a JSON-LD format. RO-Crate is the fundamental unit of uploading and downloading workflows to and from the WorkflowHub. These efforts are the basis for more domain-specific tasks, such as the provenance collection in our use case, metabolomics data, which require domain-specific standards and ontologies. 

Metabolomics is an emerging omics field that studies the small molecules from different biological samples obtained via different metabolomics techniques [16]. Metabolomics experiments yield complex data due to chemical diversity, different analytical techniques, heterogeneous laboratory instruments, and subsequent data analysis [17]. An untargeted metabolomics workflow for high-throughput data usually includes (1) the sampling and extraction of metabolites, (2) measurements of thousands of features using metabolomics instruments, (3) data processing, (4) statistical analysis, and (5) metabolite annotation and identification [18]. Each workflow step can introduce artefacts to the whole procedure and affect reproducibility [16]. 

In this case study, we demonstrated the implementation of basic guidelines and relevant metabolomics domain specifications to increase support of the FAIR principles as part of the metabolomics workflow design. We have employed a use-case workflow for metabolomics data analysis, the Metabolome Annotation Workflow (MAW) [19]. Here, we elaborate on how we made this research workflow transparent and reproducible, packaged with FAIR-supporting metadata. These guidelines are specifically tailored for workflow data objects; other data objects may require different standard guidelines [8]. This FAIR-MAW framework can be extended to more complex bioinformatics/cheminformatics workflows.

## 2. Methods: Demonstration on Making Metabolome Annotation Workflow (MAW) FAIR

With the advancements in omics technologies, there has been an exponential increase in the digital data volume [20]. The relatively new omics field, metabolomics, is the analysis of small molecules, also known as metabolites, from biological samples such as cells, tissues, or organisms, which gives insights into biological phenomena and has applications ranging from drug discovery to ecosystem monitoring [21]. The untargeted metabolomics experiments acquire thousands of features from the samples that are difficult to reproduce with different mass spectrometry instruments (described in Section 3) [22]. The FAIRification of metabolomics workflows can address some of the challenges, including data complexity and reproducibility in metabolomics, which can help researchers share their results in a more FAIR-compliant way. Here, we demonstrate the application of the FAIR guidelines to a metabolomics workflow as an illustrative example.

### 2.1. Use Case: Metabolome Annotation Workflow (MAW)

The Metabolome Annotation Workflow (MAW) [19] is a Liquid Chromatography–Tandem Mass Spectrometry (LC-MS^2^)-based metabolomics data analysis pipeline implemented in R and Python. MAW was developed as part of the Cluster of Excellence “Balance of the Microverse”, a research initiative committed to microbial communication research, which aims to implement standard practices and support FAIR data and software implementation [23]. This workflow integrates tools and packages to provide chemical structure annotation to mass spectrometry (MS) data. It takes .mzML format spectral (ontology ID—3244) data as input [24]. The final output is a CSV format file containing all spectral features and the associated structural annotations. The origin of the candidate structure is traceable through intermediate CSV/TXT files. 

MAW is divided into three components. The first component is MAW-R, which represents the R section of the workflow. It takes the .mzML LC-MS^2^ spectra files (obtained from the RAW files generated by the mass spectrometer or—for secondary analysis—available in any spectral data submission repository, such as the MetaboLights repository [25]) and three spectral databases (GNPS [26], HMDB [27,28], and MassBank [29]) stored as spectra objects in separate Robject files and available on Zenodo [30]. MAW-R generates results from spectral database dereplication as CSV files and also creates TXT parameter files for MetFrag [31] (an in silico fragmentation-based annotation tool), which are used in the second component of the workflow called MAW-MetFrag. Considering that most metabolic studies search for novel natural products, we have provided a CSV (currently hosted on Zenodo) from the COlleCtion of Open Natural ProdUcTs (COCONUT) database [32] SDF file from January 2022. MetFrag generates result CSV files for each precursor mass [*m*/*z*] with an annotated structure using the local CSV file for COCONUT, which serves as a compound database for dereplication. Alternatively, MAW also integrates SIRIUS [33], but SIRIUS 5 version requires registration, and, thus, we cannot enable sharing or reproducibility using it; hence, this study does not discuss it. The third component is MAW-Py, the workflow’s Python section. The CSV files from spectral database dereplication results and MetFrag compound database dereplication results are analysed with MAW-Py, which performs candidate selection and, for each .mzML file, generates a CSV file consisting of precursor masses [*m*/*z*] and their corresponding top 1st candidate structure. Figure 1 briefly overviews the workflow components, inputs, intermediary results, and outputs.

### 2.2. FAIRification of MAW

MAW’s first release was available for execution within a Docker container, fulfilling the minimum FAIR requirement. The recent FAIRification process for MAW is illustrated in Figure 2. For each component of the FAIR principles, specific guidelines are followed as proposed in the workflow/research software community. The FAIR principles published in 2016 [6] provided the basic concepts, and the FAIR guidelines for workflows published in 2020 [8] provided guidelines specific for workflows, which are applied to MAW and are presented in the following subsections, serving as a checklist for the FAIRified MAW. Each subsection introduces the concept of different FAIR principles, the associated rules and guidelines, and the application of these guidelines to MAW. The data and metadata linked to the workflow are also FAIR-compliant. Details of different sub-components from each FAIR letter are introduced and described by [6]: findable (F1–F4), accessible (A1–A2), interoperable (I1–I3), and reusable (R1) [7]. The FAIRification process for workflows is generally divided into two tasks: (1) the FAIR Workflow [8] and its description and (2) the FAIR execution of workflows in the context of the data [15]. In this tutorial, we focus on the workflow description aspect. It is also important to note that the demonstration of the FAIRification of MAW is not sequential but is divided into individual FAIR components. 

#### 2.2.1. Findability

The concept of findability is associated with making the research workflow findable by humans and machines. A research workflow is made findable by assigning a persistent identifier (PID), such as a Digital Object Identifier (DOI), to make the workflow unambiguously identifiable from the exact location within cyberspace. Associated metadata and keywords can be used to search for the workflow in search engines such as Google or in domain-specific registries such as WorkflowHub. A clear description of the workflow should be provided to promote the discovery and reusability of the workflow by other researchers. Findability is further distributed into four subsections.

##### Assigning a Persistent Identifier to the Workflow (F1)

Assigning a persistent identifier (PID) is the first component to make the workflow “F”indable. PIDs are unique and permanent identifiers assigned to digital objects to ensure longevity. Many workflow repositories automatically provide a PID, such as a DOI, during submission. Submitting the workflow to repositories protects the workflow from potential changes or the discontinuation of its original hosting platform. In the case of MAW, we linked the workflow code repository to WorkflowHub (details on submission below in F4), which then was used to generate a DOI (https://doi.org/10.48546/WORKFLOWHUB.WORKFLOW.510.2 (accessed on 29 January 2024)). 

##### Use Descriptive Metadata (F2)

Descriptive metadata associated with the workflow make it easier to find over different search engines and repositories, so providing as much relevant metadata as possible is recommended. The metadata can include information and contacts of developers; detailed descriptions and types of input and output data; names and versions of different tools, packages, and software integrated into the workflow; keywords; context; the software’s licence; and more. Following best practices, specific ontologies are used to represent certain digital objects consistently. For MAW, we integrated the EDAM ontology [36] to describe the .mzML format for input files, a specific format for LC-MS data. An .mzML file uses XML Schema to describe comprehensive information for a single LC-MS run, including data and metadata. This can be achieved via the CWL (details in the Section 2.2.3 Interoperability subsection). The EDAM ontology is added to the input YAML file for the CWL as the format for the .mzML file using the full URL: http://edamontology.org/format_3244 (accessed on 9 February 2024).

The descriptive metadata are also used for making the workflow findable over the “web of life sciences”. To achieve this, Bioschemas domain-specific profiles make available a semantic markup, consisting of metadata, to the web search engines specific to the life sciences domain [37,38]. Bioschemas provides community-specific types and profiles with agreed vocabularies that can be required, recommended, or optional. WorkflowHub initiated the type and profile for workflows, termed “ComputationalWorkflow”. This Bioschemas document can be generated using the examples in https://bioschemas.org/profiles/ComputationalWorkflow/1.0-RELEASE (accessed on 29 January 2024). During the submission process of MAW to WorkflowHub, the CWL description was used as a baseline to generate the RO-Crate JSON object created using the ComputationalWorkflow profile of Bioschemas.

Other resources with metadata, such as keywords, include the GitHub repositories or the published research articles. For MAW, we have used the following terms as keywords to describe the workflow in the publication: “Untargeted metabolomics”, “Workflow”, “Tandem mass spectrometry”, “FAIR”, and “Metabolite annotation”. A descriptive title and workflow summary are also required to make it more user-friendly. We used the title “Metabolome Annotation Workflow”, which describes the main task of the workflow, “to provide structural annotations to metabolomics data”. A workflow summary and a detailed description of the different functions within the workflow are described in the README.md markdown files archived on Zenodo (https://doi.org/10.5281/zenodo.8205567 (accessed on 29 January 2024)) [39].

##### Associate Workflow with Metadata Using Identifiers (F3)

Generally, the workflow and its descriptive metadata are stored in different files. It is important to mention the PID(/DOI) of the workflow in the metadata file so that the workflow is linked to the metadata. This is already present in the Bioschemas generated during the WorkflowHub submission process. In the example of MAW, the “identifier”, “https://doi.org/10.48546/WORKFLOWHUB.WORKFLOW.510.2 (accessed on 29 January 2024)”, was added to the RO-Crate JSON file written using the Bioschemas format automatically during the WorkflowHub submission process. 

##### Registering to Searchable Repositories (F4)

The first three sections on findability ensure that the digital object (workflow, in this case) has a PID, rich metadata, and a link between the two. However, all this effort is insufficient to make the workflow discoverable over the “web of life sciences”. To locate the workflow, the Bioschemas markup can be used to enhance findability in Google and other search engines, which index the Bioschemas markup. The workflow must also be submitted to a registry to be able to index it in a domain-specific webspace. WorkflowHub automates the whole findability process. In this case, MAW was submitted to WorkflowHub as a GitHub repository, which automated the F1, F2, F3, and F4 processes. 

To submit the workflow to WorkflowHub, the GitHub repository for MAW was directly linked using https://github.com/zmahnoor14/MAW.git (accessed on 29 January 2024), particularly linking the maw.cwl file to the WorkflowHub entry. WorkflowHub automatically extracted the title, description, and licence using the GitHub repository for MAW, while the keywords were added manually. After submitting this information publicly, WorkflowHub made the entry available and generated RO-Crate JSON and SVG objects (using Workflow RO-Crate profile https://about.workflowhub.eu/Workflow-RO-Crate/ (accessed on 29 January 2024)). The JSON object was written using the ComputationalWorkflow Bioschemas profile, which can be downloaded from the https://workflowhub.eu/workflows/510 entry (accessed on 29 January 2024). Once the workflow submission was executed and inspected for any error, the workflow was given a DOI using Datacite integrated with WorkflowHub, generated after the submitter approved minting a DOI to the submitted workflow version. With updates to the workflow, a new DOI can be assigned. To keep track of updates in the workflow, MAW is versioned through Github, and each release receives a DOI in Zenodo. The CWL logs record the data transformations and the parameters used, which help track the data updates/changes.

#### 2.2.2. Accessibility

The next component of the FAIR principles is the accessibility of the research workflow, ideally without any legal, technical, or financial barriers. This requires a working workflow version, readily available for download from a domain-specific repository or web interface, and licence information, even when the associated workflow is not open-access. FAIR digital objects are not always associated with open data formats or software depending on the research and the licence applied to it, so the information on authentication and authorisation should also be provided if required. 

##### The Workflow and Metadata Are Retrievable by Their Identifier Using a Standardised Communication Protocol (A1)

A1 has two essential measures. The first measure is to use a standardised protocol for the retrieval of the workflow and its metadata, which means that the user should be able to access at least the metadata using the internet and a web browser. The second measure is to provide the terms and conditions under which the workflow is accessible. This does not necessarily mean that the workflow is openly available but that the terms and conditions can be accessed for free. For MAW, the workflow metadata are accessible via WorkflowHub, which uses a standard protocol such as HTTPS (for web access) and an API (for programmatic access) to retrieve the workflow metadata. Lastly, the licence used for MAW is MIT, which allows permissive free software usage [40].

##### Metadata Should Be Accessible Even When the Workflow Is No Longer Available (A2)

Research objects disappear over time due to insufficient funding or initiative to keep them alive and running. In such cases, the metadata must have a longer availability span. The metadata maintenance is much easier and cheaper than the research object itself and can provide helpful information in case someone wants to reproduce the results using the workflow. Ensuring the longevity of metadata is also achieved by WorkflowHub through the assigned DOI.

#### 2.2.3. Interoperability

Interoperability for workflows refers to the easy exchange of information between different platforms in a standardised and compatible way. Due to layered complexity, it is considered the most challenging among the FAIR components, specifically for research software and workflows. For example, a workflow is a dynamic digital object that executes specific tasks and holds together many components, including data, tools, the execution environment, dependencies, scripts, and the source code. The workflow has various components, each of which should also be interoperable. Common workflow standards and formats should be used to make workflows interoperable, recognised by different WfMSs, and executed on different operating systems. The workflow standards solve the issues with dependencies that can be retrieved from containers, such as Docker or Singularity containers, and seamlessly allow the sharing and reuse of the workflow.

##### Workflow Uses a Standardised and Interoperable Language for Representation (I1)

To provide a common interpretation of a workflow, the same terminologies must be used throughout the workflow and its execution. So, a common standard language is required, which can be quickly learned, easily shared across different platforms, and is flexible enough to fit different workflow layouts. Common examples of such knowledge representations are JSON-LD, YAML, OWL, and RDF. These are standardised and open formats. The Common Workflow Language (CWL) is an open community workflow standard that uses YAML (or JSON) to describe workflows. It provides a common abstraction to workflows and combines different workflow components in one YAML document. The CWL can describe workflows developed using different programming languages and command line tools, facilitating portability across multiple platforms. Any workflow described using the CWL can be executed using a CWL-compliant workflow engine, such as StreamFlow [35] or *cwltool* [41]. *cwltool* acts as a workflow executor, but the execution depends on the individual tools and libraries integrated into the workflow, which requires associated containers and scripts/tools. 

The main CWL document describes workflow inputs, steps, requirements, and outputs. Depending on the type of workflow, there could be just one CWL document or more. For example, in a nested workflow, the main description can refer to the sub-workflows (/steps) in external CWL files. The names of the inputs and outputs are exchanged within the CWL description to point back to the source CWL description of these variables and to use the same data entities throughout the workflow execution. The CWL also supports the use of Docker-format software containers run by any compatible software container engine, such as Docker, Singularity, or Podman, for the portability of analysis tools. Software containers serve as a software unit with all the code and its dependencies packaged together, which can be executed in different computing environments [42]. 

Here, we used the Common Workflow Language (CWL) standards to describe our workflow and enable technical interoperability in MAW. There are three individual sub-workflows named MAW-R, MAW-MetFrag, and MAW-Py, each described using the CWL. Each step has inputs (e.g., .mzML files as inputs for MAW-R), Docker-format software container images (e.g., zmahnoor/maw-r:1.0.8), scripts (e.g., MAW-R.r) or command line tools (e.g., MetFrag), arguments, and outputs (e.g., outputs of MAW-R include CSV and TXT files). The outputs from MAW-R are used as inputs in MAW-MetFrag and MAW-Py, and MAW-MetFrag also produces outputs that are inputs for MAW-Py (Figure 1). MAW-Py then analyses the outputs of MAW-R and MAW-MetFrag to generate the final results. Data portability is supported by CWL’s feature of explicitly differentiating between ordinary strings and file paths, which enables MAW to run locally or remotely. Figure 3 illustrates the sub-workflows of MAW, together with inputs and outputs.

##### Use of FAIR Vocabularies (I2)

The individual steps and metadata of the workflow should use defined FAIR vocabularies, which can be domain-specific and should have identifiers to link them to the description of the terms. This ensures the findability and accessibility of the vocabulary used in the workflow. In the case of MAW, we employed the EDAM ontology for the .mzML file. The RO-Crate profile generated by WorkflowHub also collected the metadata using defined schemas (Bioschemas) to represent workflow-relevant provenance.

##### Linking Qualified References among Metadata (I3)

The links between different data objects are essential for the provenance collection and the contextual knowledge of different aspects of the workflow. The CWL provides these links between different entities in the description document. For example, it depicts that output from sub-workflow 1 is used as input for sub-workflow 2. These links can be more meaningful by adding contextual metadata. Another way to link the references to metadata is to use DOIs of the individual steps, tools, or input files.

#### 2.2.4. Reusability

Reusability in the FAIR principles aims to enable workflows that can be reused in multiple contexts. This includes the reproducibility of the same test data used for the development of the workflow, the reuse of the workflow for similar or dissimilar data to analyse the usage for different types of data, and, lastly, extending the functionalities of software to incorporate multiple data types and projects. Workflows can evolve and develop over time, with each component changing differently. So, enabling reusability on all levels is essential by providing detailed provenance about the workflow and its execution with different datasets. Provenance is metadata about data entities, activities or steps, and agents that generate the data using the workflow and the relational connection between research artefacts. Provenance could either be “prospective”, which captures an abstract specification of the workflow, serving as a blueprint for future reuse of the workflow or “retrospective”, which captures information about the execution of the workflow with different data and the derivation of different data entities [43]. FAIR workflows should have associated prospective provenance and, ideally, there is also retrospective provenance available for a typical usage of the workflow. The workflow should also be provided with an F/OSS licence for the subsequent users or developers to self-determine how to build upon, change, use, access, or distribute the workflow. 

##### Metadata Are Richly Described with a Plurality of Accurate and Relevant Attributes (R1)

Reusability R1 can be linked to findability category F2, where the metadata are provided for the workflow to be discovered. However, in the case of R1, it is also important to provide contextual metadata aligning with the workflow concept and its usage. These metadata can be as rich as possible, as the developers are encouraged to provide metadata from every conceivable aspect of the workflow reusability, such as the advantages and limitations, hardware requirements for execution, self-explanatory function/modules names or their descriptions, and the versions. The metadata can surpass the above-given list based on individual projects. 

Reusability (R1) refers to the following: (1) The licence associated with the workflow and the data should be mentioned in the metadata. FAIR software licences [44] include both open and closed licences, as FAIR does not imply open-source research objects. For accessibility, it is important to clearly describe the way of requesting access to the software/data objects in a standardised format. In academia, it is common for governmental funding agencies to request research objects to be open at some point. In this case, Creative Commons licences are used for datasets, and open-source licences, such as the MIT licence, Berkeley Source Distribution (BSD) licence, Apache licence, and GNU General Public License (GPL) are typical for software/workflows. In the case of MAW, we used an MIT licence for the workflow, and the associated data uploaded to Zenodo use Creative Commons Attributions 4.0 International. (2) Associated provenance for reuse of the workflow and data. MAW provides detailed prospective provenance in the GitHub README.md archived on Zenodo. The prospective provenance is also generated with the Workflow RO-Crate in an automated way during the WorkflowHub submission process. (3) Following domain relevant standards and rules to allow an easy exchange of knowledge and reuse of the digital objects. Table 1 details the different FAIR principles and the specific tools, registries, or ontologies that ensure adherence to each principle.

## 3. Results and Discussion: Towards Reproducibility in FAIR Metabolomics Data, Software, and Workflows

Metabolomics provides a global “small molecules” content within a biological sample. It is an ever-evolving analytical approach to comprehend the chemical dark space. A metabolomics experiment starts with sample preparation. The metabolites extracted from the sample are then subjected to the appropriate analytical tools such as Tandem Mass Spectrometry (MS^2^). The chemical data are acquired as spectra, which are analysed with various downstream analysis steps. The execution of each stage in the metabolomics experiment can introduce variations hindering reproducibility [16]. The replicates of the same sample under different instruments, data acquisition modes, and data processing tools can give varying results even with the same parameters [17]. Despite ongoing efforts to standardise the entire metabolomics pipeline (including experimental steps and data acquisition and processing steps), obtaining reproducible data from metabolomics experiments remains one of the biggest challenges, as it depends on the analytical aspects, such as the instrument operator, run time, etc., and biological replicates’ differences [16]. Determining the specific step in the workflow that has the most impact on reproducibility is crucial. 

The metabolomics community has defined standards for the metabolomics workflow analysis called the Metabolomics Standards Initiative (MSI) [42,43] and has recommended best practices to implement FAIR principles to metabolomics research data objects [45,46]. The MSI gives minimal standards for reporting various data types and metadata obtained from different metabolomics techniques. It also gives standards for reporting data preprocessing and metabolite identification results. However, ensuring that the standards are followed while reporting the data and their results is not enforced during metabolomics computational workflows, which makes it difficult to reuse the workflow. The reproducibility challenge in this tutorial is focused more on the computational metabolomics aspects, which are more dependent on the computational variations [47], such as the environment, parameters, execution, and minimal standards reporting. After generating the data, the first and foremost task is to submit the RAW metabolomics data files, directly obtained from the instrument, to a metabolomics repository such as MetaboLights, which adheres to minimal reporting standards for data and associated metadata [48]. The second aspect is the analysis workflow used to obtain significant results, which can be highly context-specific but general enough to be standardised for a particular job. The basic computational metabolomics workflow includes (1) data preprocessing, (2) statistical analysis, and (3) annotation. Different R, Java, and Python packages, as well as tools with command line (CLI) versions and a graphical user interface (GUI), have been developed to tackle each module of the computational metabolomics workflow [26,30,31,33,49,50,51,52]. 

In metabolomics workflow, the algorithms or parameters affect the results for (1) preprocessing and (2) statistical analysis. A fixed environment, usually within a container to run these two tasks, can be used to obtain reproducible results. The (3) annotation or identification module also depends on different spectral or compound databases that are regularly updated and can hinder reproducibility. To bypass this issue, a locally stored database/library or a fixed online version of a database (such as HMDB version 5 from https://hmdb.ca/downloads (accessed on 29 January 2024)) can be used. Each software, tool, and library should be tested individually to ensure reproducibility throughout the workflow. During the FAIRification process and reproducibility testing, it is important to acknowledge the dynamic nature and rapid evolution of the workflows [53] and sustain the maintenance and deployment of the workflow. 

Reproducibility in computational workflows can be generally enabled in two ways: the workflow is either packaged together with all its contents or shared as a recipe that can be followed to reproduce the results [8]. MAW is an automated untargeted metabolomics workflow that incorporates both ways via the CWL. The CWL specifies workflow abstractions; integrates different tools and packages; and provides a complete, deployable, and interoperable workflow description. The sharing, reuse, and archiving of the source code, installation instructions, and tutorial for MAW is achieved via Zenodo and WorkflowHub as the primary workflow registry. It is a free and open-source workflow, with all dependencies also being open-source (except for the annotation tool SIRIUS, which is integrated into another version of the workflow for interested academic users). 

The expedition to make MAW FAIR started with the open-source code policy, and though the FAIR practices were initiated during the workflow development, they persisted throughout its lifecycle and are anticipated to continue as new features are integrated into the workflow. Essential to the FAIRification process were (1) the utilisation of the CWL description and (2) the submission to WorkflowHub, both of which greatly simplified and automated the process. Comprehensive CWL description of MAW and the workflow execution via *cwltool* streamlined the workflow standardisation and execution, with the possibility to extract retrospective provenance and parallelise multiple input files. Utilising WorkflowHub’s integration of fundamental FAIR components, including Workflow RO-Crate and Bioschemas, required minimal effort from the developer side given that a GitHub repository and CWL description was already available for MAW. While the automatically generated RO-Crate during WorkflowHub submission lacked detailed workflow component explanations, this contextual description is available on Zenodo for reference. Further FAIRification of the workflow requires proper maintenance, including unit tests for all modules and dependencies, which is partially covered with Docker containers at the time.

Reproducibility is one of the main goals of FAIR principles, especially for the reuse of the workflow for different datasets. The collection of provenance is essential for reproducibility. This calls for a need for standards and ontology requirements in metabolomics studies to specify the type and format of metadata collected from the workflow as a research object (prospective provenance), along with the execution of the workflow and its results from a particular metabolomics dataset (retrospective provenance) to track the dataflow [54]. The next major phase for the FAIRification of the metabolomics workflow is to define minimum requirements/standards and ontologies for metabolomics data analysis provenance collection. Once the minimum requirements reporting for metabolomics workflows and the results obtained from these workflows are attained, the FAIR principles will be adapted to the new data objects and ontologies.

## 4. Conclusions

FAIR principles play a crucial role in advancing the reproducibility of metabolomics workflows by promoting interoperability, sharing, and reuse. The complex nature of metabolomics workflows integrating various components necessitates working towards better implementations of the FAIR principles to improve the reliability and consistency of results. This narrative demonstration exhibits an application of the FAIR principles using the Metabolome Annotation Workflow (MAW) as an example, showcasing the role of WorkflowHub as a FAIR workflow registry and the utilisation of the Common Workflow Language (CWL) to enable standardised practices and interoperability across different platforms. By embracing FAIR principles and establishing minimum standards and ontologies for metabolomics workflows and the associated provenance, we can increase reproducibility, facilitate effective data sharing, improve interoperability by reducing the dependency on specific workflow engines or languages, and foster further advancements in computational metabolomics.

## Figures and Tables

**Figure 1 metabolites-14-00118-f001:**
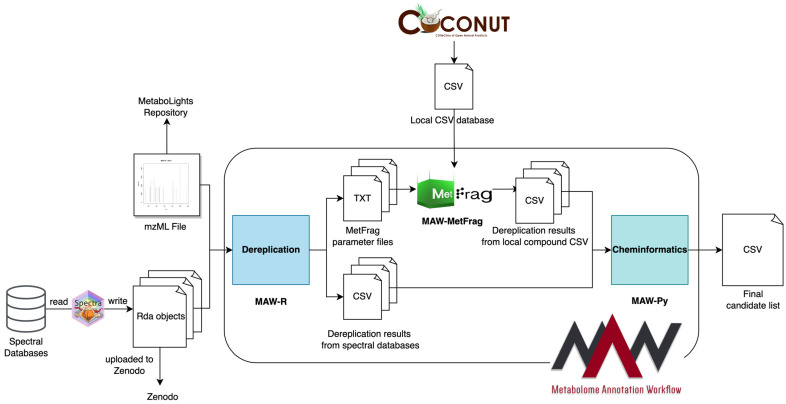
Overview of three MAW components. MAW is divided into three sections termed MAW-R (spectral database dereplication), MAW-MetFrag (compound database dereplication; here, as an example, we provide the COCONUT database), and MAW-Py (cheminformatics post-processing and candidate selection). The inputs for the first component, MAW-R, are .mzML input files, which can be uploaded to MetaboLights. The spectral databases (GNPS, HMDB, and MassBank) are downloaded and written as spectra R objects and then uploaded to Zenodo. These inputs are used to run MAW-R, which generates CSV results from spectral database dereplication and TXT files for MetFrag parameters. MAW-MetFrag, the second component, takes these input TXT files and generates CSV outputs from compound database dereplication. Once we have results from the first two components, MAW-Py, which is the third component, takes the CSV files from MAW-R and MAW-MetFrag to select candidate structures and post-process the results to give a final list of precursor masses with their corresponding top candidate structure.

**Figure 2 metabolites-14-00118-f002:**
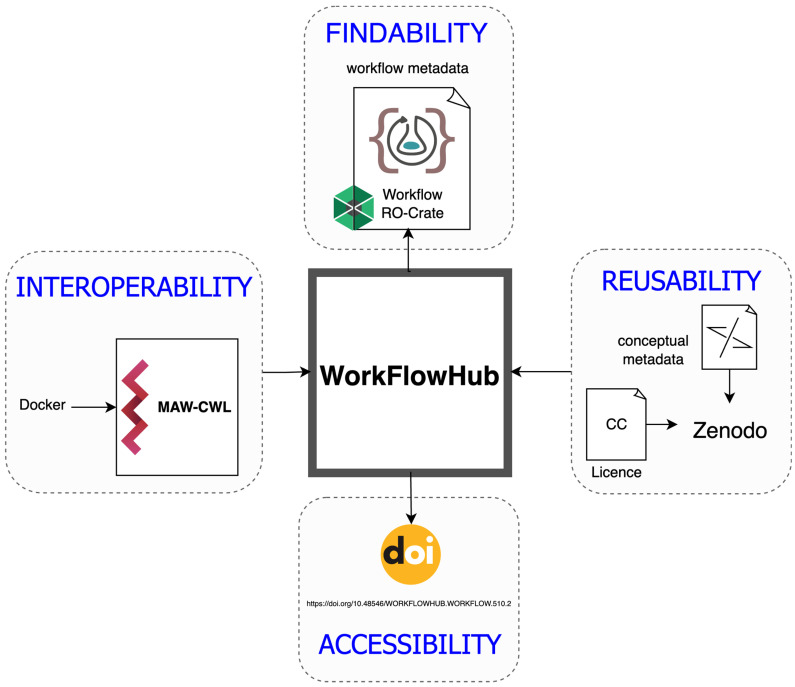
FAIRification scheme for MAW. WorkflowHub, a workflow registry, is the foundation to enable FAIR-supporting workflows. Metabolome Annotation Workflow (MAW) is described using Common Workflow Language (CWL), which supports the FAIRification process and is WorkflowHub’s preferred format. MAW is findable; the workflow metadata are packaged as a workflow RO-Crate object, which includes Bioschemas markup to enable its findability across the life sciences web. MAW is accessible and archived with WorkflowHub versioning system linked with a DOI. MAW is interoperable; it is described using CWL and employs Docker-format software container images to set up a deployable environment across different systems. *cwltool*, the reference implementation of CWL, can also act as a workflow runner, as well as other workflow engines that support CWL, such as Toil [34] and Streamflow [35]. MAW is reusable; the contextual metadata and tutorial on the usage of MAW are archived on Zenodo, with the source code and a software licence. MAW’s interoperability and reusability aspects were defined first and submitted to the WorkflowHub, enabling findability and accessibility.

**Figure 3 metabolites-14-00118-f003:**
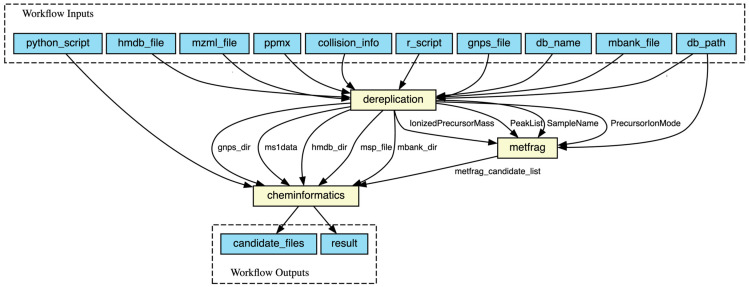
CWL description of MAW workflow and its sub-workflows for the execution of a single .mzML file. The environment setup with all the dependencies is provided via Docker-format software container images, one defined for each sub-workflow of MAW (dereplication—MAW-R, MetFrag—MAW-MetFrag, and cheminformatics—MAW-Py). The first section in the figure showcases all the workflow inputs, including the R and Python scripts, the input .mzML file, the paths to spectral databases (hmdb_file, gnps_file, mbank_file), the information on collision energy used to fragment the molecules (collision_info), the relative mass deviation in parts per million (ppm), and finally the compound database name and its path (db_name, db_path, which is the COCONUT database). The dereplication step (MAW-R) takes all inputs except the Python script. Once dereplication is executed, MetFrag takes the parameters generated from dereplication step, such as IonizedPrecursorMass, PeakList, SampleName, and PrecursorIonMode. Once MetFrag step is also executed, cheminformatics (MAW-Py) step takes outputs from dereplication (gnps_dir, ms1data, hmdb_dir, msp_file, mbank_dir) and MetFrag steps (MetFrag_candidate_list) and generates final outputs, the candidate files (candidates lists for each molecule), and results (final top 1st candidate list for all molecules). The inputs are provided via a YAML file. This whole workflow can be scattered over multiple .mzML input files for parallelisation.

**Table 1 metabolites-14-00118-t001:** FAIR supporting tools, ontologies, and registries and the respective FAIR principles.

FAIR Components	Docker	CWL	Bioschemas	RO-Crate	WorkflowHub
Findable	✓		✓	✓	✓
Accessible	✓				✓
Interoperable	✓	✓	✓	✓	✓
Reusable	✓	✓	✓	✓	✓

## Data Availability

MAW is available with its source code, installation instructions, and tutorial on GitHub (https://github.com/zmahnoor14/MAW (accessed on 29 January 2024)) [55]. It is registered as a workflow on WorkflowHub with DOI 10.48546/WORKFLOWHUB.WORKFLOW.510.1. The Docker images for MAW are https://hub.docker.com/repository/docker/zmahnoor/maw-r/general (accessed on 29 January 2024), https://hub.docker.com/repository/docker/zmahnoor/maw-metfrag_2.5.0/general (accessed on 29 January 2024), and https://hub.docker.com/repository/docker/zmahnoor/maw-py/general (accessed on 29 January 2024). The COCONUT database is archived on Zenodo with the DOI 10.5281/zenodo.7704937 (accessed on 29 January 2024) [56]. Spectral databases are archived on Zenodo with the DOI 10.5281/zenodo.7519270 (accessed on 29 January 2024) [57]. Small benchmark data and a workflow execution are available on the MAW GitHub repository (https://github.com/zmahnoor14/MAW/blob/main/cwl/Usage_Example.md (accessed on 29 January 2024)) [58].

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
