# Peer review of "Implementation of FAIR Practices in Computational Metabolomics Workflows—A Case Study"

_metabolites, 2024, doi:10.3390/metabo14020118_

Round 1

Reviewer 1 Report

Comments and Suggestions for Authors

This is a nicely written manuscript about implementing FAIR practices in the development of computational metabolomics workflows. They use the workflow they developed, Metabolite Annotation Workflow (MAW)  as a case study to show how each of the FAIR principles are met. Overall a very nice paper that will be useful to others. I only have a few comments 

1) The ultimate goal is for reproducible research and it seems that the authors have done a great job with MAW to meet the FAIR principles. It would be good if they could demonstrate with a real data set (maybe with two different users on different computational systems or other way) how the goal of the FAIR principles were met. This could go in a supplement but a real data example could be a strong demonstration.

2) Licensing is mentioned. Please elaborate on the types of license that one may use in computational metabolomics. This may be helpful for readers as they develop their workflows.

3) Please discuss more about how updates are handled while still adhering to FAIR principles. It is mentioned a few times but a separate section with some more detail would be helpful. 

4) Please provide any new references more specific to metabolomics. For examples there are some articles regarding FAIR principles and databases PMID: 37482620, or best practices more generally for software development: PMID: 33467846 

Minor:

1) There are two Figure 2s

Author Response

We would like to thank the reviewer for the pertinent comments and helpful suggestions. These really enriched the manuscript. Below, please find the responses point-by-point, together with the changes that have been made to the manuscript.

Reviewer 1

This is a nicely written manuscript about implementing FAIR practices in the development of computational metabolomics workflows. They use the workflow they developed, Metabolite Annotation Workflow (MAW)  as a case study to show how each of the FAIR principles are met. Overall a very nice paper that will be useful to others. I only have a few comments.

Major: 

  • The ultimate goal is for reproducible research and it seems that the authors have done a great job with MAW to meet the FAIR principles. It would be good if they could demonstrate with a real data set (maybe with two different users on different computational systems or other ways) how the goal of the FAIR principles was met. This could go in a supplement, but a real data example could be a strong demonstration.
    1. The Demonstration is now available on: https://github.com/zmahnoor14/MAW/blob/main/cwl/Usage_Example.md and is mentioned in the manuscript's Data Availability section.

  • Licensing is mentioned. Please elaborate on the types of license that one may use in computational metabolomics. This may be helpful for readers as they develop their workflows.
    1. We have added the following lines in the manuscript to describe the licences.
      1. Line 435 - 443: “FAIR software licences [44] include both open and closed licences, as FAIR does not imply open-source research objects. For accessibility, it is important to clearly describe the way of requesting access to the software/ data objects in a standardised format. In academia, it is common for governmental funding agencies to request research objects to be open at some point. In this case, Creative Commons licences are used for datasets, and open-source licences MIT, Berkeley Source Distribution (BSD), Apache, GNU General Public License (GPL) are typical for software/workflows. In the case of MAW, we used an MIT licence for the workflow, and the associated data uploaded to Zenodo uses Creative Commons Attributions 4.0 International.”

  • Please discuss more about how updates are handled while still adhering to FAIR principles. It is mentioned a few times, but a separate section with some more detail would be helpful. 
      1. Since tracking/handling updates is obtained via GitHub and Zenodo archiving which we already described in the context of Findability and Accessibility, we have not dedicated a separate section to updates, However, for more clarity we have added the following text to the manuscript.
        1. Line 287 - 289: “To keep track of updates in the workflow, MAW is versioned through Github, and each release gets a DOI in Zenodo. CWL logs record the data transformations and the parameters used, which help track the data updates/changes.”
  • Please provide any new references more specific to metabolomics. For examples there are some articles regarding FAIR principles and databases PMID: 37482620, or best practices more generally for software development: PMID: 33467846 
    1. The following lines are now added to the manuscript.
      1. Line 481 - 484: “The metabolomics community has defined standards for the metabolomics workflow analysis called Metabolomics Standards Initiative (MSI) [42, 43], and have recommended best practices to implement FAIR principles to metabolomics research data objects [44, 45].”

Minor:

  • There are two Figure 2s
    1. Fixed

Reviewer 2 Report

Comments and Suggestions for Authors

In this study, as far as I understand, a case study was conducted that takes FAIR principles into account. However, information should also be provided about whether the previously developed software package is scalable, maintainable, and testable, or more precisely, whether it was developed with a clean architecture approach.

Author Response

We would like to thank the reviewer for the pertinent comments and helpful suggestions. These really enriched the manuscript. Below, please find the responses point-by-point, together with the changes that have been made to the manuscript.

  1. In this study, as far as I understand, a case study was conducted that takes FAIR principles into account. However, information should also be provided about whether the previously developed software package is scalable, maintainable (versioning/ fixing), and testable, or more precisely, whether it was developed with a clean architecture approach.
    1. The clean architecture approach was not the focus of the work, but we see the potential benefits and could explore this in future work.
    2. The initial version of the workflow followed minimum FAIR criteria. We have added the following lines to the draft about the previous version of the workflow. MAW’s first release fulfilled minimum FAIR requirements: Code publicly available on GitHub, archived on Zenodo with the DOI: 10.5281/zenodo.7148450. The workflow was only available for execution within a docker container, which already fulfils most of the FAIR criteria. I have added following lines to the manuscript:
      1. Line 171- 172: “MAW’s first release was available for execution within a docker container, fulfilling minimum FAIR requirement.” 
    3. The current release with CWL improved the scalability of the workflow as different highly scalable workflow management systems support CWL (like Toil and Streamflow) and consequently can execute MAW. 
    4. For maintainability, the docker images are published on DockerHub for each step. An ideal approach for future development would be to use unit tests for different modules of the workflow. We have added the following lines to the discussion section of the manuscript.
      1. Line 541 - 543: “Further FAIRification of the workflow requires proper maintenance, including unit tests for all modules and dependencies, which is partially covered with docker containers at the time.”

Reviewer 3 Report

Comments and Suggestions for Authors

Major comments

Whilst the manuscript is well-written, I feel it would benefit from clearly stating (Abstract, Conclusions) who the target audience is, and what the target audience is intended to do with the manuscript. At present, the Abstract references using "the instructions presented in this snapshot as a base template" and the Conclusion references "This tutorial..." But where are the instructions, and where is the tutorial, for someone conducting a metabolomics workflow for the first time? I believe the authors should decide whether this is a narrative discussion, or a set of tutorials and instructions for practitioners, and make sure that the manuscript reflects this choice. For example, if it is a tutorial with a set of instructions, these should be logically reproducible by the reader and applicable to a real-world case study.

As a secondary major comment, I am very unclear what additional / novel information is included within this paper, versus the author's very recent paper from just a few months ago,
"MAW: the reproducible Metabolome Annotation Workflow for untargeted tandem mass spectrometry:"
https://doi.org/10.1186/s13321-023-00695-y

Specifically, the previous paper, which I have read closely, already deals with  how the workflow contributes "Towards a FAIR and reproducible MAW", so can the authors provide a clear statement about what is novel versus the previous paper, which already states that "For good scientifc practice, in MAW, we implemented the recommended FAIR principles"?

Especially, I was surprised to see that the previous paper is only referenced twice (lines 96 and 116). I think the authors should be much more clear about what is novel in this manuscript, as compared with their previous published paper, in order to demonstrate that the new work is significant and not just an extended "methods" section describing how they implemented the recommended FAIR principles.

Minor points

Line 123, the url inserted in the middle of the sentence is not that helpful to readability, I would include it as a reference rather than the middle of the sentence, or maybe at the end.

Line 128, Does the workflow also work for files generated by instruments? If so, it might be better written as "The first component is MAW-R, which represents the R section of the workflow. It takes the .mzML LC-MS2 spectra files (generated by the mass spectrometer, or - for secondary analysis - available in any spectral data submission repository such as MetaboLights"

Line 144 and elsewhere, m/z should be italicised

Line 413, typo in "Resubility"

Line 439, "Nucleic Magnetic Resonance (NMR)" should be Nuclear Magnetic Resonance. Why mention NMR here, unless I misunderstood this workflow is only applicable to mass spectrometry? If you are just making a general observation that NMR can produce metabolomics data, this feels like a comment that belongs in Introduction, its is not a Result or Discussion.

Why no Limitations discussion, for example the challenges of using a system such as MAW in more novel settings or matrices? 

What will the authors report under "Data Availability Statement: to add."

Author Response

We would like to thank the reviewer for the pertinent comments and helpful suggestions. These really enriched the manuscript. Below, please find the responses point-by-point, together with the changes that have been made to the manuscript.

Major comments

  • Whilst the manuscript is well-written, I feel it would benefit from clearly stating (Abstract, Conclusions) who the target audience is, and what the target audience is intended to do with the manuscript. At present, the Abstract references using "the instructions presented in this snapshot as a base template" and the Conclusion references "This tutorial..." But where are the instructions, and where is the tutorial, for someone conducting a metabolomics workflow for the first time? I believe the authors should decide whether this is a narrative discussion, or a set of tutorials and instructions for practitioners, and make sure that the manuscript reflects this choice. For example, if it is a tutorial with a set of instructions, these should be logically reproducible by the reader and applicable to a real-world case study.
      1. It is more of an experience report which can be adopted by other metabolomics workflow developers, which is now more clear in the abstract and conclusion.
        1. Line 27 - 29: “Researchers can use this narrative discussion as a guideline to commence using FAIR practices for their bioinformatics or cheminformatics workflows while incorporating necessary amendments specific to their research area.”
        2. Line 564 - 565: “This narrative demonstration exhibits an application of the FAIR principles using the Metabolome Annotation Workflow (MAW) as an example”
  • As a secondary major comment, I am very unclear what additional / novel information is included within this paper, versus the author's very recent paper from just a few months ago, "MAW: the reproducible Metabolome Annotation Workflow for untargeted tandem mass spectrometry:"
    https://doi.org/10.1186/s13321-023-00695-y Specifically, the previous paper, which I have read closely, already deals with  how the workflow contributes "Towards a FAIR and reproducible MAW", so can the authors provide a clear statement about what is novel versus the previous paper, which already states that "For good scientific practice, in MAW, we implemented the recommended FAIR principles"? Especially, I was surprised to see that the previous paper is only referenced twice (lines 96 and 116). I think the authors should be much more clear about what is novel in this manuscript, as compared with their previous published paper, in order to demonstrate that the new work is significant and not just an extended "methods" section describing how they implemented the recommended FAIR principles.
    1. MAW’s first release fulfilled the minimum FAIR requirement: Code publicly available on GitHub, archived on Zenodo with the DOI: 10.5281/zenodo.7148450. The workflow w, which already fulfils most of the FAIR criteria. I have added the following lines to the manuscript:
      1. Line 171- 172: “MAW’s first release was available for execution within a docker container, fulfilling minimum FAIR requirement.” 
    2. The original workflow paper is about the workflow from an application point of view with analysis. In this manuscript, we focused only on the work for adopting the practices recommended in the FAIR principles to the workflow.

Minor points

  • Line 123, the url inserted in the middle of the sentence is not that helpful to readability, I would include it as a reference rather than the middle of the sentence, or maybe at the end.
      1. Added the ID instead of the url as reference
  • Line 128, Does the workflow also work for files generated by instruments? If so, it might be better written as "The first component is MAW-R, which represents the R section of the workflow. It takes the .mzML LC-MS2 spectra files (generated by the mass spectrometer, or - for secondary analysis - available in any spectral data submission repository such as MetaboLights"
      1. The following statement has now been added to the manuscript: 
        1. Line 134 - 138: “It takes the .mzML LC-MS2 spectra files (obtained from the RAW files generated by the mass spectrometer, or - for secondary analysis - available in any spectral data submission repository such as MetaboLights repository [25])  and three spectral databases (GNPS [26], HMDB [27, 28], and MassBank [29]) stored as Spectra objects in separate Robject files and available on Zenodo [30]. ”
      2. The MS generates RAW files, which can be converted to mzML using any open-source software such as GNPS or ProteoWizard. 
  • Line 144 and elsewhere, m/z should be italicised
      1. Fixed
  • Line 413, typo in "Resubility"
      1. Fixed
  • Line 439, "Nucleic Magnetic Resonance (NMR)" should be Nuclear Magnetic Resonance. Why mention NMR here, unless I misunderstood this workflow is only applicable to mass spectrometry? If you are just making a general observation that NMR can produce metabolomics data, this feels like a comment that belongs in Introduction, its is not a Result or Discussion.
      1. I have now removed the Nuclear Magnetic Resonance (NMR)  word from the Discussion, as the technique is irrelevant to the workflow.
  • Why is there no Limitations discussion, for example, the challenges of using a system such as MAW in more novel settings or matrices? 
      1. We talked about Provenance in the discussion with the following text as a future development task, which MAW currently lacks:
        1. Line 546 - 550: “This calls for a need for standards and ontologies requirements in metabolomics studies to specify the type and format of metadata collected from the workflow as a research object (prospective provenance), but also the execution of the workflow and its results from a particular metabolomics dataset (retrospective provenance) to track the dataflow [51].”
      2. Additionally, we added the following text to mention the lack of workflow testing as a further limitation.
        1. Line 541 - 543: “Further FAIRification of the workflow requires proper maintenance including unit tests for all modules and dependencies.”
  • What will the authors report under "Data Availability Statement: to add.
    1. The following text has been added to the Data Availability Statement:
      1. Line 587 - 596: “MAW is available with its source code, installation instructions, and tutorial on GitHub (https://github.com/zmahnoor14/MAW) [55]. It is registered as a workflow on WorkflowHub with DOI: 10.48546/WORKFLOWHUB.WORKFLOW.510.1. 
        1. The docker images for MAW are https://hub.docker.com/repository/docker/zmahnoor/maw-r/general, https://hub.docker.com/repository/docker/zmahnoor/maw-metfrag_2.5.0/general, and https://hub.docker.com/repository/docker/zmahnoor/maw-py/general. The COCONUT database is archived on Zenodo with the DOI: 10.5281/zenodo.7704937 [56]. Spectral databases are archived on Zenodo with the DOI: 10.5281/zenodo.7519270 [57]. A small benchmark data and workflow execution are available on MAW GitHub repository (https://github.com/zmahnoor14/MAW/blob/main/cwl/Usage_Example.md) [58].”

Round 2

Reviewer 1 Report

Comments and Suggestions for Authors

The comments were adequately addressed.

Reviewer 3 Report

Comments and Suggestions for Authors

The authors have responded appropriately to my questions and improved the manuscript. Thank you.